# Reproducibility report for ML Reproducibility Challenge 2022

**Scope of Reproducibility**

The goal of this replication study is to find out to what extend the results of the paper "Robust Counterfactual Explanations on Graph Neural Networks" [1] are reproducable. We investigate this matter by evaluating the following claims:

- Using the trained RCExplainer and baseline RCExplainer-NoLDB, does it deliver the same performance on fidelity as stated in the paper?
- Using the trained RCExplainer and baseline RCExplainer-NoLDB, does it deliver the same performance on robustness as stated in the paper?

**Methodology**

In order to reproduce the results as stated in the original paper of Bajaj et al. [2021] [1], we used the originally provided source code. However, contrary to our expectations, the provided source code on its own was not enough to redo the experiments. Therefore, the main approach of this reproducibility paper was to adjust the provided source code in order to execute some of the the conducted experiments. The source code and our adjustments are written in Python and used the PyTorch library.

**Results**

The first claim in the scope of reproducibility was not accepted in terms of this paper. The RCExplainer was close to the results found by manual training but outside of the range provided by the standard deviation of manual training. The performance of the RCExp-NoLDB model showed a major difference from the results reported in the paper. The robustness results were hard to compare due to lack of actual numbers, but qualitative analysis found it to be reproducible, as values lied within margin of the standard deviation of manually trained models.

**What was easy**

The original paper [1] was very understandably written which made it very accessible. As a result, our vision towards the conducted experiments and their execution became easily clear to us. The straight forward vision of the paper made it easy to understand what we were aiming to reproduce.

**What was difficult**

As the source code was provided, we implied that it would be trouble-free to run the experiments and evaluate the results. However, we quickly found out that this was not the case. The source code contained a large number of code files containing of various bugs, such as; duplicate functions, missing arguments when calling functions and missing files. This and the large number of code lines made it difficult to debug.

**Communication with original authors**

When investigating the source code we encountered some difficulties, which made us reach out to the original authors via email. In here, we asked for clarification on the RCExplainer without linear decision boundaries (No-LDB), as we doubted our obtained results. Unfortunately, the authors did not come back to us.

## 1 Introduction

Graph Neural Networks (GNNs) have shown to be of great success in the field of be graph representation learning [4]. GNNs exploit the structure of graph data by integrating node information in terms of position and its neighbouring nodes [2]. The proven potential of GNNs call for better explainability of the models, which are otherwise considered as black boxes [9]. Numerous attempts in GNN explainability have been made [8, 5, 6]. However, Bajaj et al. [2021] [1] argue that these methods lack in robustness and counterfactuality. Therefore, they proposed their paper "Robust Counterfactual Explanations on Graph Neural Networks" [1], which attempts to tackle these problems.

The paper proposes the RCExplainer, which is a novel method to produce robust and counterfactual explanations on GNNs. The RCExplainer attempts to model the decision logic of a GNN, by modelling its decision regions using linear decision boundaries (LDBs). The paper explores the common decision logic encoded in those boundsaries, so that they are able to produce counterfactual and robust explanations. The key concept of the RCExplainer are the proposed decision regions derived by the LDBs. In order to demonstrate the effectiveness of this method, the authors also proposed a RCExplainer-NoLDB (no linear decision boundaries). The RCExplainer-NoLDB follows the same framework as the RCExplainer, however it did not use the LDBs to generate explanations.

The paper tests the RCExplainer and RCExplainer-NoLDB against three other state-of-the-art GNN explainers (GN-NExplainer [8], PGExplainer [5] and PGM-Explainer [6]). The paper conducted two main experiments. One of them focused on counterfactuality (measured in fidelity), whereas the other experiment focused on robustness. The results of their experiments show that the RCExplainer defeats all other mentioned explainers in terms of fidelity and robustness. In addition to this, the RCExplainer's efficiency is either the same or outperforms the efficiency of the other explainers. This paper will investigate these results in means of a replication study.

## 2 Scope of reproducibility

The focus of this replication study is on the reproducibility of the experiments conducted by Bajaj et al. [2021] [1]. More specifically, we will focus on reproducing the results of the RCExplainer and RCExplainer-NoLDB. The effectiveness of the aforementioned models is tested in two experiments, which are elaborated on in section 4. The original paper conducted more experiments which are presented in their appendix. However, due to time limitations, this reproducibility paper will only focus on the experiments as presented in the result section of their paper.

The paper publicly offers the original code used to implement the RCExplainer and RCExplainer-NoLDB. Using this code, we are testing the reproducibility of the paper by firstly training both models from scratch. Additionally, we will use the source code in order to evaluate both models. Both models will be evaluated by the same means as mentioned in their paper, so that the circumstances will be the same as in their presented experiments. Our paper focusses on accepting/rejecting the following statements:

1. Using the trained RCExplainer and RCExplainer-NoLDB, does it deliver the same performance on fidelity as stated in the paper?
2. Using the trained RCExplainer and RCExplainer-NoLDB, does it deliver the same performance on robustness as stated in the paper?

In the following sections we will first discuss the RCExplainers and RCExplainer-NoLDBs workings, after which the experimental setup needed to run the tests checking the reproducibility of the paper. Following, the methodology will be discussed in which the methods used to get the code provided in the paper to work are mentioned. Finally, we will discuss the results of the tests and whether these support the paper. Thereafter, a discussion is provided to look back upon the process of producing this reproducibility report.

## 3 RCExplainer

The authors start the construction of the RCExplainer by modelling decision regions. The decision regions are described as convex polytopes, which are formed by Linear Decision Boundaries (LDBs). In order to satisfy the desired fidelity and robustness the polytopes are required to have the following properties:

1. The decision region should be induced by a subset of the LDBs in $\mathcal{H}$. Here, $\mathcal{H}$ is the set of LDBs induced by $\phi_{fc}$, and $\phi_{fc}$ is the mapping function that maps the finalized graph embeddings to a predicted distribution over the classes.

2. The decision region should cover many graph instances in the training dataset D, and all the covered graphs should be predicted as the same class.

The first property enables the counterfactual explanations from the LDBs to correspond with the real decision logic of the GNN. The second property ensures the decision region to capture the common decision logic on all graphs covered by the decision region, i.e. the explainer will not overfit on singular data inputs. The decision regions are extracted by optimizing the following equation:

$$\max_{\mathcal{P} \subseteq \mathcal{H}} g(\mathcal{P}, c), \text{ s.t } h(\mathcal{P}, c) = 0, \tag{1}$$

in which $\mathcal{P}$ is a subset of LDBs in $\mathcal{H}$, g($\mathcal{P}$, c) is the number of graphs covered by $\mathcal{P}$ that are in class $c$, and h($\mathcal{P}$, c) is the number of graphs covered by $\mathcal{P}$ that are **not** in class $c$. The $\mathcal{P} \subseteq \mathcal{H}$ satisfies the first property. In addition to this, the maximization of g($\mathcal{P}$, c) while keeping h($\mathcal{P}$, c) zero satisfies the second property.

Following, the decision regions were used to train a model $f_\phi$ that produces the desired counterfactual and robust explanations on input graphs. Say we have input graph $\mathcal{G}$ with two connected vertices $v_i$ and $v_j$. We get the following equation:

$$\mathbf{M}_{ij} = f_\phi(\mathbf{z}_i, \mathbf{z}_j) \tag{2}$$

Where $\mathbf{z}_i$ and $\mathbf{z}_j$ are the graph embeddings of $v_i$ and $v_j$, produced by the last convolutional layer of a GNN and $\mathbf{M}_{ij}$ denotes the probability of the vertice connecting $v_i$ and $v_j$ belonging to the explanation. For an input graph with n vertices, $\mathbf{M} \in \mathbf{R}^{n \times n}$. The produced explanation for each input graph is defined by selecting all vertices where $\mathbf{M}_{ij} >$ 0.5.

Finally, the loss function that is used to train the RCExplainer is defined in equation 3.

$$\mathcal{L}(\theta) = \sum_{G \in D} \{\lambda \mathcal{L}_{same}(\theta, G) + (1 - \lambda)\mathcal{L}_{opp}(\theta, G) + \beta \mathcal{R}_{sparse}(\theta, G) + \mu \mathcal{R}_{discrete}(\theta, G)\} \tag{3}$$

Here, $\lambda \in [0, 1], \beta \geq 0$ and $\mu \geq 0$ are hyperparameters, each controling the importance of the reported term. For the full definition of each loss term, see appendix A.0.1.

## 3.1 RCExplainer-NoLDB

As previously described in the introduction, the authors of the paper proposed the RCExplainer-NoLDB in order to prove the effectiveness of using LDBs to define decision regions. The RCExp-NoLDB uses the same framework as the RCExplainer, but excludes the LDBs from generating the graph explanations. In contrast to the RCExplainer, the RCExplainer-NoLDB solely focuses on maximizing the prediction confidence. Formally said; the RCExplainer-NoLDB directly maximizes the prediction confidence on class $c$ for $G_\phi$ and minimizes the prediction confidence of class $c$ for $G'_\phi$.

The difference in maximization problem calls for a change in loss function. The RCExplainer-NoLDB uses the same loss function as the GNNExplainer and the PGExplainer. However, the loss of the RCExplainer-NoLDB introduces a second term, and is defined in equation 4.

$$\mathcal{L}_{conf}(\theta, G) = -log(P_\phi(Y = c|X = G_\theta)) - \frac{\eta}{log(P_\phi(Y = c|X = G'_\phi))} \tag{4}$$

Here, $P_\phi(Y|X = G_x)$ defines the conditional probability distribution learnt by the GNN model $\phi$ for input graph $G_x$. Y represents the random variable of the set of classes $\mathcal{C}$, and $\mathcal{X}$ is the random variable representing possible input graphs for the GNN $\phi$. The newly introduced loss term is minimalized together with the $\mathcal{R}_{sparse}$ and $\mathcal{R}_{discrete}$ term as described in section 3 and appendix A.0.1.

## 4 Experimental setup

The following section will discuss the experimental setup as proposed by the authors of the RCExplainer. As explained in the introduction, this reproducibility paper will focuss on two of the experiments as conducted in the original paper;

one testing the counterfacuality and one testing the robustness of their explainer. The reported experiment test the fidelity and robustness of the generated explanations on the task of graph-classifications (note: the original appendix also includes experiments on singular node-classification, however this will not be touched in the scope of this paper).

## 4.1   Counterfactuality

The first experiment tested the degree of counterfactuality of the RCExplainer against their own RCExplainer-NoLDB, the GNNExplainer [8], the PGExplainer [5] and the PGM-Explainer [6]. As a measure of counterfactuality, the authors used fidelity. They define it as "The drop in confidence of the original predicted class, after masking the produced explanation in the original graph". In other words, fidelity measures the confidence of the prediction after removing a set of edges from the input graph. The formal definition of fidelity for input graph $G$ and explanation $S$ is given in equation 5.

$$fidelity(S, G) = P_\phi(Y = c | X = G) - P_\phi(Y = c | X = G_{E \setminus S}) \tag{5}$$

Here, $c$ denotes the class predicted by $\phi$. The fidelity was plotted against sparsity. Sparsity is defined as the percentage of edges remaining after the explanation is removed from $\mathcal{G}$. Equation 6 shows the formal definition of the sparsity of explanation $S$ w.r.t input graph $G = (V, E)$.

$$sparsity(S, G) = 1 - \frac{|S|}{|E|} \tag{6}$$

## 4.2   Robustness performance

The second experiment proposed by the authors focused on the robustness of their RCExplainer. E.g. how well their explainer performed while adding noise to the input graph. Noise was added to the input graph $G$ by randomly deleting edges or adding random noise to the node features. This was all done under the requirement that the prediction on the noisy $G$' is consistent with the prediction on $G$. As a measure of robustness they used the Area Under the Curve (AUC) of the receiving operator characteristic (ROC). Their paper accepted the top 8 edges of $S$ as the ground truth and compared these results to noisy $S$'. The robustness performance experiment excluded the PGM-explainer, as it did not output the necessary values needed to complete the experiment.

## 4.3   Datasets

The reported results made use of one synthetic dataset: BA-2Motifs [5], and two real world datasets: Mutagenicity [3] and NCI1 [7]. However, due to the limited timespan and the extensive hours it takes to train the models, our replication study will only reproduce the results based on the Mutagenicity dataset. The Mutagenicity dataset consists of 4337 datapoints, in which each entry is a specific molecule. Every datapoint carried a label of being either mutagenic or non-mutagenic.

## 5   Methodology; re-implementation of code

The open source code contained a notebook which loaded the RCExplainer that was pre-trained on the Mutagenicity dataset. In addition to this, the notebook contained all the necessary code to evaluate the RCExplainer on fidelity and visualize the explanations. Furthermore, the function to train the RCExplainer from scratch was included. However, when running the notebook, we had issues with certain libraries not being compatible with each other (despite using the correct environment). In addition to this, we found some other flaws causing errors. Eventually, with some adaptions, we got the notebook to run fluently. However, the notebook was limited to specific datasets, models and evaluation tasks. In order to conduct the robustness experiment, we had to look into the source code. The provided source code was written in python 3.8, whereas the models used the PyTorch extension.

The source code consisted of a numerous amount of (long) code files. The README file contained an explanation on how to properly run the code, e.g. what commands were needed to evaluate and train the models. However, it swiftly became clear that their provided instructions on how to run the code did not work as intended. This was due to the fact that, for example, the source code contained many out-commented lines of code that were crucial for the task. In

addition to this, some of their provided code lines were inherently incorrect. Due to the many files and lacking structure, it was difficult to debug the code.

As a result of this, our expected methodology changed during the process of re-implementation. Ultimately, our approach consisted of understanding and debugging the files in order to train the models and run the desired experiments.

The following alterations were made to enable the source code to run the experiments:

- The training function as provided in the notebook was set to 0 epochs. We changed the arguments supplied to training method to 600 epochs.
- Directories in supplied code were hardcoded to use creators file structure. These were changed to work on multiple different machines using relative paths instead of absolute paths.
- Running code to evaluate models as described in README was not possible due to missing files (i.e. Mutagenicity_gt_edge_labels_new.p). The code to create those files were integrated within the training process of a GNN (other than the desired explainers). However, this code was commented out and thus the neccesary files were missing.
- To train this GNN, a different version of PyTorch and CUDA was needed than the ones supplied in the environment. Therefore, the environment had to be updated.
- Parameters for evaluating and training models were made dynamic instead of being reset to default every time, making it impossible to run different configurations.
- Invalid function calls were removed on objects (i.e. .item() on a list object).
- Missing parameters were added in function calls (i.e. stats.update(masked_adj, imp_nodes) $\rightarrow$ stats.update(masked_adj, imp_nodes, sub_adj, sub_feat, sub_nodes)) to prevent errors.
- The default parameter "bloss_mode" was changed to sigmoid to prevent error: bloss_version= "sigmoid".
- The saving of tensors was changed to the saving of tensor values. This solved a "CUDA: out of memory error" due to pytorch being unable to garbage collect thousands of tensors saved in a list.
- Function calls of non existing functions were changed to existing functions (i.e. addEdges2() $\rightarrow$ addEdges()).
- Missing parameters that were not used in their respective functions were removed.

For sake of brevity we have excluded changes made that are duplicates of or variations on the changes above.

## 6 Results

The following section will either accept or reject the following claims as stated in section 2:

- Using the trained RCExplainer and RCExplainer-NoLDB (as a baseline), does it deliver the same performance on fidelity as stated in the paper?
- Using the trained RCExplainer and RCExplainer-NoLDB (as a baseline), does it deliver the same performance on robustness as stated in the paper?

First of all we attempted to run the provided notebook properly. Setting up the correct environment was quite troubling on most of the devices. For example, the notebook demanded extra lines of code to make it run, as different versions of the torch, torchvision and torchaudio packages (other than specified in the requirements) had to be installed.

By default, the notebook was structured around the task of evaluating the pre-trained RCExplainer on the fidelity task. The notebook contained a specific section on how to train the RCExplainer from scratch. However, it took some effort to discover how to set the correct parameters to properly activate the training process. Hereafter, an error occurred in which the explainer returned NoneType. The train function was wrongfully called, as train() is a void function.

When resolved, it was possible to train the RCExplainer from scratch in the notebook. Training the RCExplainer-NoLDB from scratch had to be done through the source code. This involved alternations to the provided code. A list of these alternations is stated in section 5. Although the notebook allowed us to train the RCExplainer, we also investigated whether training from source code was possible. However, as explained in section 5, following the README instructions did not immediately allow for success.

To conclude, it is possible to train the RCExplainer and RCExplainer-NoLDB from scratch. However, it is important to notice that it was quite challenging to fulfill this task. The provided source code, including the README file, created

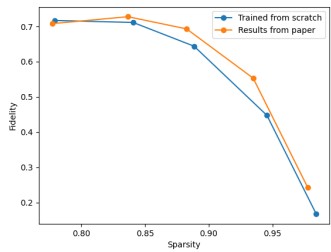 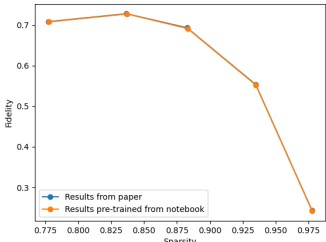

(a) Fidelity results of the paper vs (b) Fidelity results of the paper vs pre-
trained from scratch                          trained model

Figure 1: Fidelity results of several models

the false illusion that the code could run with just the correct commands. Howbeit, we managed to reach the goal of training the models.

## 6.1 Result 1

First of all, we conducted a smaller sub-experiment. The source code included one of their pre-trained RCExplainers (seed not specified). We ran the fidelity evaluation on this model and obtained the results as shown in figure 1b. The results show that their pre-trained model delivers almost identical results as reported in the paper. This seems questionable considering our results as described in the next paragraph. However the scope of this reproducibility paper is not wide enough to draw conclusions from this observation.

Subsequently, we started to investigate the first claim being made in this paper; "Using the trained RCExplainer and RCExplainer-NoLDB, does it deliver the same performance on fidelity as stated in the paper?". In order to properly evaluate both models, we trained the models 3 times using different seeds. The default code in the notebook was used to run the fidelity experiment on both models. The final result as reported in table 1 contain the averaged fidelity and the corresponding standard deviation.

Table 1: fidelity results for the RCExplainer and RCExplainer-NoLDB trained from scratch on Mutagenicity dataset

(a) RCExplainer

| Sparsity | Fidelity | STD |
|---|---|---|
| 78 % | 0.72 | 0.009 |
| 84 % | 0.71 | 0.002 |
| 88 % | 0.64 | 0.014 |
| 95 % | 0.45 | 0.019 |
| 98 % | 0.17 | 0.007 |
| 100 % | 0.00 | 0.000 |

(b) RCExplainer-NoLDB

| Sparsity | Fidelity | STD |
|---|---|---|
| 75 % | 0.05 | 0.015 |
| 80 % | 0.04 | 0.010 |
| 85 % | 0.03 | 0.006 |
| 90 % | 0.02 | 0.003 |
| 95 % | 0.01 | 0.002 |
| 100 % | 0.00 | 0.000 |

Running the experiment on fidelity requires the calculation of sparsity within. Therefore, our results differ in sparsity from the results reported in the original paper. The difference in x-values makes it hard to statistically test the similarities. The results for the RCExplainer are plotted in figure 1a. From here we can conclude that both results show very similar trends. However, when looking at table 1a, our reported standard deviations are quite small. Especially for sparsity between 0.85% and 0.95%, the difference in our results and the reported results exceed our reported standard deviation. We can therefore conclude that the difference in results is too large to accept the claim. Nonetheless, this could be the result of the use of different seeds, or the difference in number of trained models (10 vs 3).

The results of the experiment conducted with the RCExplainer-NoLDB is shown in figure 2. The provided source code did not include the numbered values of their RCExplainer-NoLDB, hence we can only compare the charts in figure 2a and 2b. Their source code also provided us with one pre-trained RCExplainer-NoLDB model (no seed specified), which's result are plotted in 2a. Immediately visible is the difference between our trained model trained and their results. From this we started questioning if our training process ran properly. However, as later described in section 6.2, our models did perform great on the robustness experiment. The robustness also showed big improvements when

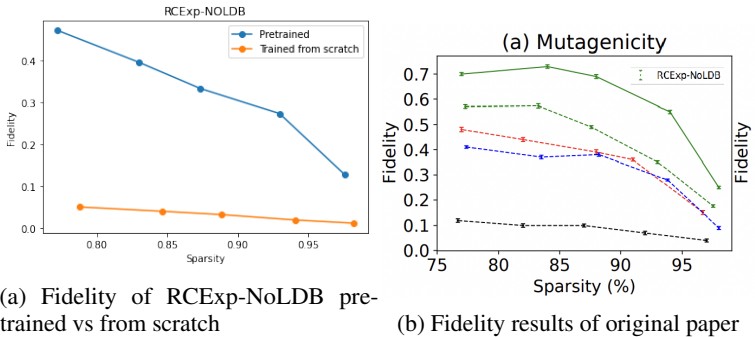

(a) Fidelity of RCExp-NoLDB pretrained vs from scratch

(b) Fidelity results of original paper

Figure 2: Fidelity results of several models

comparing the results before and after training, confirming that the model did learn. We were not able to find the cause of the massive divergence in results.

Hence, we can conclude that we were unable to reproduce the results of the RCExp-NoLDB. However, the difference in results is so significant that this was most likely caused by an error in the way the model was trained. The training was done using the supplied code without any major adjustments. Furthermore, we used the same evaluation techniques as for the RCExplainer. Changes made to the code were of similar scope to the changes described in section 5. Therefore we had no indication on what was causing the divergence in results. Another noticeable result is the poor performance of the pre-trained RCExp-NoLDB. As the fidelity in both our performed research as well as in the original paper comes with a small standard deviation, it is unlikely that the pre-trained model contributed to the reported fidelity results.

To conclude, investigating the averaged results produced by the models trained from scratch, our RCExplainer results seem to be close to the ones reported in the paper. Our numbers are slightly lower than the ones found in the paper, and are not within close enough margin to support the papers claims. In addition to this, we did not manage to reproduce the RCExplainer-NoLDB results.

## 6.2 Result 2

The final claim in investigation is: "Using the trained RCExplainer and RCExplainer-NoLDB, does it deliver the same performance on robustness as stated in the paper?". We used the same trained models as mentioned in the previous section, whereas the reported results are again the averaged performance. Contrary to the previous experiment, it was not possible to change the settings of the notebook to robustness evaluation. Therefore, we had to make more adjustments in the source code. The README file stated that the experiment could be run by adding '–noise' to a specific console command. Following these exact instructions, did not lead to successfully running the experiment, as parameters entered in the console were overwritten in the code. Besides, the code responsible for calculating the AUC contained many errors, among which missing parameters in function-calls and out-commented crucial code. For more information, see section 5. Ultimately the experiments succeeded, the results are shown in table 2.

Table 2: Robustness results for the RCExplainer and RCExplainer-NoLDB trained from scratch on Mutagenicity

(a) RCExplainer

| Noise | AUC | STD |
|-------|------|-------|
| 0 % | 1.00 | 0.000 |
| 5 % | 0.98 | 0.001 |
| 10 % | 0.94 | 0.001 |
| 15 % | 0.91 | 0.001 |
| 20 % | 0.88 | 0.003 |
| 25 % | 0.85 | 0.004 |
| 30 % | 0.83 | 0.004 |

(b) RCExplainer-NoLDB

| Noise | AUC | STD |
|-------|------|-------|
| 0 % | 1.00 | 0.000 |
| 5 % | 0.98 | 0.001 |
| 10 % | 0.94 | 0.000 |
| 15 % | 0.91 | 0.001 |
| 20 % | 0.87 | 0.000 |
| 25 % | 0.85 | 0.007 |
| 30 % | 0.82 | 0.004 |

The paper did not report the actual values of their robustness experiment. Therefore, the conclusion we draw are statistically hard to support, as actual values are missing. We compared our results in figure 3a and 3b and see that the results are very similar. However, the chart as reported in the paper does not provide high enough precision to correctly

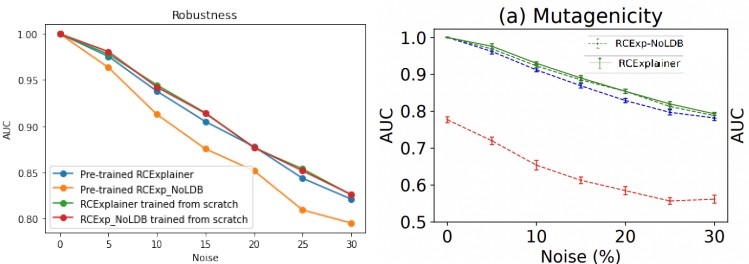

(a) Robustness of several models      (b) Robustness results of original paper

Figure 3: Robustness results of several models

draw a conclusion from it. Therefore, we also conducted the sub-experiment of evaluating their provided pre-trained model on robustness. As figure 3a displays, all of the results were very similar. Based on this observation, we conclude that our results are similar within the range of acceptance.

# 7 Discussion

As can be derived from section 6, it was hard to draw clear conclusions on the claims as set in 2. Respectively, we rejected the first claim and accepted the second claim. The first claim was rejected as their results did not fall within the scope of the standard deviation. However, our results were based on 3 model runs, in comparison with the original paper which used 10 model runs. The deviation in results could be due to this reason, which should be looked into further. In addition to this, more time to investigate the failures as stated in section 6.1 could possibly lead to the correct reproduction of the experiment. Therefore, our rejection of the first claim might be unfair, and could be looked into in future research. Finally, the limited time for this project restricted us from conducting the experiments as proposed in their appendix. The appendix contained valuable experiments and results, worth looking into. Consequently, our conclusion can not render a full statement on the reproducibility of their entire paper.

## 7.1 What was easy

Bajaj et al. [2021] [1] wrote a very straight forward paper. It was easy to dive into the literature of their paper, and we had a very clear vision on the purpose and set-up of their research. In addition to this, once we got the code to run properly, it was very easy to conduct the said experiments. The technical aspect of their code worked fluently. We did not have to rewrite any technical aspect of the models, neither did we have to adjust code in the experiments.

## 7.2 What was difficult

As previously stated, it was hard to properly test the results of the paper for reproducibility on our local machines. The original idea of this research was to run all experiments as stated in the paper on all datasets. In addition to that, we originally thrived to set-up a new experiment to test whether the models also perform outside the scope of their paper. However, the implementation of the two experiments required substantially more time than expected. Therefore, we did not manage to research the models and/or experiments outside the scope of their research.

Furthermore, we would have liked to also take the GNNExplainer into consideration as an additional baseline. However, after some minor alterations in the code (similar to the ones described in section 5), we discovered that in order to evaluate/train the GNNExplainer adversarial data was required, which was not supplied with the code. This made it impossible for us to use the GNNExplainer as a baseline as described in the paper. Besides that, we attempted to implement the original code of the GNNExplainer as described in the paper of Ying et al. [8]. However their evaluation of results was very different to what we needed in order to use it for our experiments, which again, made it not possible to use it as a baseline during this research.

Finally, the results that we managed to produce were very hard to analyse. As neither the paper or the source code contained the absolute values of the experiments, we could only examine our results by comparing charts. As the charts were relatively large-scaled, proper conclusions on similarities could not be made.

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

# A Appendices

### A.0.1 Loss function

The following equations further define the loss function as described in equation 3. In all following equations, the following notations are used:

- $\mathcal{H}_G$ = the set of LDBs that induce the decision region covering $\mathcal{G}$

- $|\mathcal{H}_G|$ = the number of LDBs in $\mathcal{H}_G$

- $\mathcal{B}_i(\mathbf{x}) = \mathbf{x}_i^T + b_i$ for the $i$-th LDB $h_i \in \mathcal{H}_G$, where $\mathbf{w}_i$ and $b_i$ are the basis and bias of $h_i$, and $\mathbf{x} \in O^d$ is a point in space $\mathcal{O}^d$.

- $\mathcal{B}_i(\mathbf{x}$ = indication of whether point $\mathbf{x}$ lies on the positive or negative side of $h_i$.

- $|\mathcal{B}_i(\mathbf{x})|$ = the absolute value proportional to the distance of point $\mathbf{x}$ from $h_i$.

- $\sigma$ = the sigmoid function

$$\mathcal{L}_{same}(\theta, G) = \frac{1}{|\mathcal{H}_G|} \sum_{h_i \in \mathcal{H}_G} \sigma(-\mathcal{B}_i(\phi_{gc}(G)) * \mathcal{B}_i(\phi_{gc}(G_\theta))) \tag{7}$$

The loss term as shown in equation 7 encourages the graph embeddings $\phi_{gc}(G)$ and $\phi_{gc}(G_\theta)$ to lie on the same side of every LDB in $\mathcal{H}_G$.

$$\mathcal{L}_{opp}(\theta, G) = \min_{h_i \in \mathcal{H}_G} \sigma(\mathcal{B}_i(\phi_{gc}(G)) * \mathcal{B}_i(\phi_{gc}(G'_\theta))) \tag{8}$$

$\mathcal{L}_{opp}(\theta, G)$ (equation 8) optimizes the counterfactuality of the explanations. The term requires the prediction on $\mathcal{G}'_\theta$ to be of significant difference of the prediction on G. It encourages the graph embeddings $\phi_{gc}(G)$ and $\phi_{gc}(G'_\theta)$ to lie on opposites side of (at least) one LDB in $\mathcal{H}_\mathcal{G}$.

$$\mathcal{R}_{sparse}(\theta, G) = ||\mathbf{M}_1|| \tag{9}$$

$\mathcal{R}_{sparse}(\theta, G)$, as shows in equation 10, is used as a L1 regularization, such that only a small number of edges in $\mathcal{G}$ are selected as the counterfactuality explanation. Matrix $\mathbf{M}$ is produced by $f_\theta$ on an input graph $G$.

$$\mathcal{R}_{discrete}(\theta, G) = -\frac{1}{|\mathbf{M}|} \sum_{i,j} (\mathbf{M}_{ij} log(\mathbf{M}_{ij}) + (1 - \mathbf{M}_{ij}) log(1 - \mathbf{M}_{ij}) \tag{10}$$

Equation 10 shows the final term of the loss function, and is also referred to as the entropy regularization. The function makes sure to push each value entry in $\mathbf{M}_{ij}$ to be close to either 0 or 1, such that $\mathcal{G}_\theta$ and $\mathcal{G}'_\theta$ approximate $G_S$ and $G_{ES}$ quite well.

