# OpenReview forum: "Reproducibility report for ML Reproducibility Challenge 2022"
_ML_Reproducibility_Challenge/2021/Fall — Reject_

### Meta-Review · Area_Chair_9212 · 2022-04-07

**Recommendation:** Reject
**Confidence:** 5

**Metareview:**

While the paper did not receive any reviews, I (the Meta reviewer) have evaluated it carefully. The paper presents good reproduction effort for the claims of the original paper. However, in certain aspects the paper lacks in presentation, writing style and grammar. Having said so, I commend the efforts of the authors to reproduce the claims even using a poor codebase from the original paper (they did not receive any communication from the original authors). It was a tough decision to reject this paper, but l believe future readers will gain valuable insights from this work.

---

### Decision · Program_Chairs · 2022-04-09

Reject